# Personalized Prediction of Patient Radiation Exposure for Therapy of Urolithiasis: An Application and Comparison of Six Machine Learning Algorithms

**DOI:** 10.3390/jpm13040643

**Published:** 2023-04-07

**Authors:** Clemens Huettenbrink, Wolfgang Hitzl, Florian Distler, Jascha Ell, Josefin Ammon, Sascha Pahernik

**Affiliations:** 1Department of Urology, Nuremberg General Hospital, Paracelsus Medical University, 90419 Nuremberg, Germany; florian.distler@klinikum-nuernberg.de (F.D.); jascha.ell@klinikum-nuernberg.de (J.E.); sascha.pahernik@klinikum-nuernberg.de (S.P.); 2Team Biostatistics and Publication of Clinical, Research and Innovation Management (RIM), Trial Studies, Paracelsus Medical University, 5020 Salzburg, Austria; wolfgang.hitzl@pmu.ac.at; 3Department of Ophthalmology and Optometry, Paracelsus Medical University Salzburg, 5020 Salzburg, Austria; 4Research Program Experimental Ophthalmology and Glaucoma Research, Paracelsus Medical University, 5020 Salzburg, Austria; 5Institute of Medical Physics, Nuremberg General Hospital, Paracelsus Medical University, 90419 Nuremberg, Germany; josefin.ammon@klinikum-nuernberg.de

**Keywords:** urolithiasis, radiation risk, awareness, personalized, neural networks

## Abstract

The prediction of radiation exposure is an important tool for the choice of therapy modality and becomes, as a component of patient-informed consent, increasingly important for both surgeon and patient. The final goal is the implementation of a trained and tested machine learning model in a real-time computer system allowing the surgeon and patient to better assess patient’s personal radiation risk. In summary, 995 patients with ureterorenoscopy over a period from May 2016 to December 2019 were included. According to the suggestions based on actual literature evidence, dose area product (DAP) was categorized into ‘low doses’ ≤ 2.8 Gy·cm^2^ and ‘high doses’ > 2.8 Gy·cm^2^ for ureterorenoscopy (URS). To forecast the level of radiation exposure during treatment, six different machine learning models were trained, and 10-fold crossvalidated and their model performances evaluated in training and independent test samples. The negative predictive value for low DAP during ureterorenoscopy was 94% (95% CI: 92–96%). Factors influencing the radiation exposure were: age (*p* = 0.0002), gender (*p* = 0.011), weight (*p* < 0.0001), stone size (*p* < 0.000001), surgeon experience (*p* = 0.039), number of stones (*p* = 0.0007), stone density (*p* = 0.023), use of flexible endoscope (*p* < 0.0001) and preoperative stone position (*p* < 0.00001). The machine learning algorithm identified a subgroup of patients of 81% of the total sample, for which highly accurate predictions (94%) were possible allowing the surgeon to assess patient’s personal radiation risk. Patients without prediction (19%), the medical expert can make decisions as usual. Next step will be the implementation of the trained model in real-time computer systems for clinical decision processes in daily practice.

## 1. Introduction

Urinary stones are one of the most common diseases in modern society. The last decades saw an increase in urolithiasis in industrialized countries [1]. The European and national guidelines recommend endourological stone removal as the major treatment option of urolithiasis [2,3]. Depending on the stone size and location, this can be performed as ureterorenoscopy (URS) or percutaneous nephrolithotomy. In both cases, radiographic imaging is important to perform effective stone treatment. The fluoroscopy with and without contrast fluid, which is used in these procedures, serves for a better guidance, control of the stone position, detection of anomalies, evaluation of complications in case of perforation and helps to place and control the position of ureteral catheters. This results in a significant risk for increased radiation exposure from the imaging and fluoroscopy used during treatment for patients with urolithiasis [4]. There is little to no transparency for surgeons and patients as to when the risk for intervention-related radiation exposure is increased. Especially the individual radiation exposure, which depends on several factors, can be nearly impossible to estimate. Patient-related factors, such as age, gender, body size and weight, as well as the characteristics of the stone to be treated (location, size, density), can have an influence on the exposure. The amount of radiation exposure for patients undergoing endourologic procedures varies based on the type and duration of the procedure. Since the disease of urolithiasis has a high probability of recurrence, subsequent interventions with increasing radiation doses are possible. There are various arguments to perform certain interventions without fluoroscopy, if they are considered to be an uncomplicated stone removal. This leads, accordingly, to no radiation exposure [5,6,7]. However, this procedure depends on the centre performing the procedure. For example, high-volume clinics with appropriate experience can easily avoid more fluoroscopy for semirigid ureterorenoscopies which may result in simple stone removal. For more complex stone removal or necessary ureteral catheter insertions/removals or control of intraoperative complications, fluoroscopy may be necessary. In recent years, the number of ureterorenoscopies steadily increased [8]. Ureterorenoscopy is one of the most important methods of stone removal due to the advancement in the field of laser lithotripsy and the standardized approach. With an increase in stone disease, ureterorenoscopy remains the most important treatment option. Considering the increase in endourological stone therapy in recent years, driven by miniaturization of instruments, the type of therapy and the associated radiation exposure is becoming increasingly relevant. Guidelines of interventional radiology already describe an individual risk of patients in dependence of the procedure. In addition to this, it is recommended to discuss the risk associated with the interventional procedure with the patient during consent process [9]. However, in daily clinical practice, the patient’s awareness about imaging modalities and radiation dose during intervention is low [10,11]. Accordingly, there are recommendations to educate patients about their exposure to radiation [12]. Studies already address radiation exposure during endourological procedures for urologic staff and their awareness [13]. In addition, factors influencing risk for higher radiation exposure during endourological therapy were identified [14]. However, there is currently no assessment tool predicting radiation exposure for the planned procedure individually, in particular for the high-risk radiation exposure patients. For endourologic stone treatment, a neuronal network model, after being trained by a machine learning algorithm and being validated for surgery planning to support the patient and the surgeon to assess the therapy planning, is already available [15].

From a methodological point of view, this paper analyses and tests various candidate predictor variables such as age of the patient, stone height, depth, width and so on in a first step of analysis. This step of univariable variable screening is carried out in many clinical studies as it is important to identify significant predictors which help the clinician to better understand a disease or event. Additionally, it reduces the complexity of the model-building process to end up with a good machine learning model. However, even if a set of univariable significant predictors was found, still the questions remain: ‘how can we combine these factors to a multivariable set of multivariable variables in order to predict the outcome on a personalized level: in our case, whether dose area product (DAP) is low or high for a given patient?’, ‘Eif we have identified a set of variables with sufficient univariable predictive power, are some of them correlated within each other or somehow redundant?’, ‘which subsets of predictors are the smallest with sufficient predictive power?’, ‘which importance should be given to which predictor and what is the combined effect of several predictors?’. These are important questions as good feature selection results in simplification of models, avoids curse of dimensionality, makes models more easier to be interpreted by researcher, decreases over-learning and reduces training times.

In the past, statisticians and mathematicians provided various models and methods such as logistic regression, general discriminant analysis and various techniques for variable screening such as backward variable selection algorithms to find answers to the above questions. These methods already provided reasonable models to combine various predictors to end up with ‘good’ models. However, over the last few decades, more flexible models with better performances were found, including support vector machines, neural networks, gradient boosted trees, nearest neighbour models, Bayes classifiers and so on, within the frame of ‘supervised learning’ in machine learning theory [16].

Within this new framework, many problems and solutions in this field were recognized (e.g., better methods for feature selection such as genetic algorithms, avoiding of ‘over-learning/overfitting’ by *n*-fold crossvalidation, early stopping techniques, L2-regulation techniques and validating the model in independent test samples, ‘dimensionality reduction’, application of a ‘reject option’ to increase negative and positive predictive power at the cost that not every patient receives a prediction) and a bulk of methods and models were found in order to cope with these problems [17]. All these new methods resulted in machine learning algorithms and deep learning analysis which were finally successfully applied in various subdisciplines in urology and nephrology [18,19,20,21].

In this study, we aimed to develop, train and test six machine learning methods and to compare their performances to finally obtain a model which can be applied on a personalized level, i.e., directly to each patient allowing the surgeon and patient to better assess patient’s personal radiation risk. The final goal is the implementation of a trained and tested machine learning model in a real-time computer system which will be applied in daily practice.

## 2. Materials and Methods

### 2.1. Compliance with Ethical Standards

The study adhered to the principles outlined in the Declaration of Helsinki, and its conduct was reviewed and approved by the institutional review board (IRB-2022-19).

### 2.2. Study Design 

This Is a Single-Centre, Observational, Retrospective Clinical Study.

### 2.3. Ureterorenoscopy Technique

To be eligible for inclusion in the study, patients needed to have undergone computed tomography (CT) imaging either as part of their initial emergency presentation or at the time of their preoperative presentation. All patients who underwent URS were included. Patients with extracorporeal shock wave therapy or percutaneous nephrolithotomy were not included. Patients with anatomical anomalies such as horseshoe kidney or ureter duplex were excluded from the study. With respect to stone size, no preselection was made to accurately reflect clinical reality. The objective of the study was to provide a preoperative estimation of the levels of radiation exposure that would be incurred during stone treatment. In order to be able to classify and assess risk of exposure, threshold values for the intervention were necessary. Based on threshold values for urological interventions, which were evaluated in the large multicentre “FLASH” study, we assessed and classified whether there was an increased risk of radiation exposure for each intervention [22]. Accordingly, for ureterorenoscopy a cut-off value was set at 2.8 Gy·cm^2^ (dose area product/DAP), i.e., low dose corresponds to ≤2.8 Gy·cm^2^ and higher dose to >2.8 Gy·cm^2^. The interventions of all surgeons of the institution were included into the analysis in order to reflect as accurately as possible the daily clinical routine and the level of training of the surgeons. In order to make the models as applicable as possible in daily clinical practice, only predictors that could be easily and consistently collected and measured were included in the prediction models. Data from CT imaging were analysed to determine the predictive power of stone size, volume, density and location, as well as general patient information such as age, size, weight and medical history regarding ureteral stenting. In the majority of cases, preoperative ureteral stenting was performed during the patient’s initial emergency visit, typically due to therapy-resistant colic or a septic condition. All semirigid ureteroscopy procedures were carried out using Wolf^®^ semirigid ureterorenoscopes (Knittlingen, Germany), which ranged in size from 4.8 to 8.4 Charrière. Flexible URS were performed using 9.5 Charr. LithoVue (Boston Scientific^®^, Marlborough, MA, USA), 9.2 Charr. UScope PU3022A (Pusen^®^-Urotech, Rohrdorf, Germany) and 8.5 Charr. Axis (Dornier^®^, Friedrichshafen, Germany). Laser lithotripsy was performed using the Lumenis^®^ Pulse 120H holmium laser with a wavelength of 532 nm, manufactured in Yokneam, Israel. The standardized fragmentation program at 20W was used for the lithotripsy. A nitinol basket was used to remove the stone. The X-ray workstation was the Uroskop Omnia Max workplaces system from Siemens Healthineers, Germany.

### 2.4. Statistical Methods

Data were cleaned for inaccurate or missing data resulting in data of 827 patients with full records of each predictor as well as the outcome variable. Randomization tests based on equal and unequal variance homogeneity (5000 Monte Carlo simulations) were used to test continuously distributed data and for discrete variables, Fisher’s exact test and Pearson’s chi-square tests were used. Gradient boosted trees, support vector machines, nearest neighbours classifiers, random forest models, Bayes classifiers and multilayer perceptron neural networks were applied and their model performances were compared. The full sample (*n* = 827) was randomly split into a training sample (*n* = 521), validation sample (*n* = 57) and test sample (*n* = 249) (Appendix A). Appendix A provides a depiction of the neural network architecture, which includes details about all layers and activation functions. A 10-fold cross-validation was used for model training using 10% of training data as validation set each. To mitigate overfitting, early stopping approach using the validation sample and 10-fold cross-validation were used. Additionally, L2-regularization techniques were also used. A total of 84% of all patients had a dose area product ≤2.8 Gy·cm^2^ and this information was introduced as prior distribution for all models. Fifteen candidate predictors, which were known to have high or moderate prediction power, were selected based on prior medical expert knowledge and the results of published studies. To reduce the number of predictor variables further, a genetic algorithm for feature selection was employed [23]. Due to the categorization of the endpoint, it was obvious that no model will be able to perfectly separate low and high DAP perfectly. Thus, a ‘reject option’ was applied, i.e., instead of one cut-off, two cut-offs were applied to the a posteriori probabilities, allowing the models to reject to make a prediction [24]. Although not every patient received a prediction, this approach significantly improved the accuracy of the model performance, which allowed medical experts to rely on it with high confidence. To analyse the generalizability of the algorithms to new, previously unseen data, negative and positive predictive values (NPV and PPV) and the percentage of subjects without a prediction were computed in both the training and independent test samples. This externally verified how well the algorithms performed when confronted with new data [24]. Statistical significance for all reported tests was determined using two-sided tests and the significance level was set to 5%. The statistical analyses presented in this report were conducted using STATISTICA 13 (Hill, T. & Lewicki, P. Statistics: Methods and Applications. StatSoft, Tulsa, OK) and MATHEMATICA 13.0 (Wolfram Research, Inc., Mathematica, Version 13.0, Champaign, IL, USA, 2021) [25,26].

## 3. Results

In summary, 995 patients were enrolled into the study between May 2016 and December 2019 and 168 patients had to be excluded due to incomplete data. Finally, complete data from 827 patients were included for data analysis (Table 1).

The following parameters were found to be significant predictors: patient age (*p* = 0.0002), gender (*p* = 0.011), patient’s weight (*p* < 0.0001), ureteral stone size (height, width, depth, all *p* < 0.000001) (Figure 1).

Further significant parameters were surgeon experience (level of specialization) (*p* = 0.039), number of stones (*p* = 0.0007), mean stone density in Hounsfield units (HU) (*p* = 0.023) and maximal stone density (HUmax) (*p* = 0.005), use of flexible URS (*p* < 0.0001). Stone location was an important predictor (*p* < 0.00001). The presence of a urinary stone in the upper part of the urinary tract resulted in increased dose area products, while distal stones usually resulted in a lower dose area product (Figure 2).

An illustration of how close patients with a DAP ≤ 2.8 Gy·cm^2^ and DAP > 2.8 Gy·cm^2^ were closely stuck together is given in Figure 3. These large areas of overlap illustrate the urgent need for using the reject option, as it was unlikely that a machine learning algorithm will be able to correctly allocate patients in the overlap to the correct DAP group (low or high).

Before the application of the reject option, support vector machines showed the most promising results (Table 2) at the first look with a NPV of 90% and PPV of 89%. However, a more detailed analysis of the model performances after application of various reject options showed that all models other than the Bayes classifier were quite similar. We finally decided to work with neural networks, as these models are most convenient to implement in a real-time computer system as compared to the other algorithms.

After the application of the reject option, the best neuronal network model achieved a NPV of 94% but rejected 157 of 827 (19%) patients in the overall sample (Table 3).

For the vast majority of 670 cases (81%), a prediction was made with a corresponding negative predictive value of 94% (631/670). As no patient was predicted to have a DAP > 2.8 Gy·cm^2^, the positive predictive value is not given (Table 3 and Appendix A).

## 4. Discussion

Patients with urolithiasis have a high probability of being exposed to radiation from imaging and fluoroscopy during their treatment [4]. Technological advances such as the miniaturization of instruments and the use of effective lasers such as the holmium or thulium laser for lithotripsy allow more and more complex stone diseases to be treated endourologically. With better complication management and improved awareness of pressure and temperature levels on the kidney during endourologic surgery, effective stone treatment is possible for an increasing number of stone patients through ureterorenoscopy and laserlithotripsy. Considering the increasing number of stone patients and, thus, endourologic stone treatment, radiation exposure is an important issue. The influencing factors and their impact on radiation duration and intensity vary widely. For the patient, it is an important aspect when bringing clarity to an informed consent discussion. Radiation exposure as part of the informed consent process is not only recommended by medical experts, but patients are also becoming more self-aware and are demanding information about the expected treatment procedures and side effects. With the developed prediction model, highly accurate predictions with a negative predictive value of 94% can be achieved for most patients with ureterorenoscopy (81%), allowing the surgeon to identify patients with low personal radiation risk. The remaining cases (19%) are still the most challenging patients and predictions can be made by the medical expert as usual. This subgroup of patients should thoroughly be analysed to find new predictors with enough power to make better predictions for these patients. For patients, the use of our model allows them to better assess their personal radiation risk and to be aware that they do not have an increased radiation exposure. This is a useful asset, especially for patients with recurrent stones. The addition of this information in the patient-informed consent before surgery increased the quality of the informed consent process and addressed the side effects due to radiation exposure of endourological stone treatment.

Knowledge of the individual radiation exposure is not only important for the patient, but also for the treating surgeon who will inform the patient and discuss with the patient possible therapy alternatives. The most effective and safest type of treatment should always be chosen. However, especially in the treatment of urolithiasis, significant improvements were achieved through developments and miniaturization in the field of endourology leading to several different treatment options [2]. For patients, therefore, in addition to effective treatment, weighing risks and complications play an increasingly important role when deciding between the different treatment alternatives. Another approach is performing endourological stone removal without fluoroscopy. This technique was also shown to be feasible even for PCNL without increased risk to patients or inferior stone free rates [27,28]. However, this is not applicable to all patients and in many centres, the use of fluoroscopy is part of the stone treatment and complication management. For PCNL the stone treatment is usually expected with a higher radiation exposure [29]. Although the development of a prediction model would be useful, the number of patients undergoing PCNL is generally lower than URS patients. Additionally, it is more difficult to find appropriate reference values, as these significantly differ depending on the expertise of the performing centre [22]. In addition, there are different conditions due to the type of positioning and technique of the PCNL, which lead to different radiation exposures. Thus, the supine position has an advantage over the prone position with regard to radiation exposure [30]. This decision must be made by the surgeon performing the procedure which might enlarge the variations of DAP for PCNL. Currently, most patients are not adequately informed about the radiation risk. Thus, despite numerous recommendations, it is difficult to inform patients about their personal risk of radiation without preinterventional data. By using the model, patients’ knowledge of their radiation exposure can be improved and valuable information can be provided about the planned intervention for the stone treatment. By applying this model, the surgeon is able to classify the radiation exposure in the context of the stone treatment. This is particularly interesting for those patients who require multiple endourologic therapies for recurrent urolithiasis. For example, the decision to perform repeated flexible URS for complex stone treatment may also be influenced by the radiation risk or should be discussed with the patient. Treatment concepts are evolving more towards individualized therapy and patients differ significantly among themselves due to their risk factors. Based on the significant parameters determined by us, risk factors can be much better taken into consideration. Using neural networks, the model we proposed can help patients better assess their own radiation risk, which plays an important role in patient awareness and therapy acceptance. Importantly, radiation exposure for patients can be reduced by various technical approaches such as use of intermittent pulse rates which can reduce the radiation dose. These approaches could be used in high risk patients to reduce the individual radiation exposure. In addition, specific training protocols increase clinical knowledge for low radiation exposure but do not significantly alter fluoroscopy time [31]. The greatest challenge is to define which threshold values should be set as the limit for an intervention, since the principle of “as low as reasonably achievable” should always apply. The study by Simson et. al, published in 2020, allowed us to specify threshold values for the different procedures in order to provide guidance to patients and surgeons. Combining these reference levels with our model, it is now possible to identify a subset of 81% of all patients, of which 94% obtain a correct prediction of receiving a low DAP. It should be critically noted that the authors underline the difference of high volume and low volume hospitals and their influence on their results. This makes it challenging to set generally defined reference limits, as the expertise and resulting radiation exposure varies with the number of cases treated [22]. The final entrance skin dose of the patient was not evaluated. Since the data were derived from the dose area product and the fluoroscopy time, the actual entry skin dose and its distribution over relevant areas of the body was not recorded. This issue needs to be further investigated when it comes to the important question of physical radiation exposure for the patient.

It would be desirable to directly predict the DAP or entrance skin doses, and we suspect that additional predictors are needed for doing so. If these predictors can be gathered in clinical use, the model with these new predictors should be further analysed for their prediction power and should be used in new machine learning models.

The search for new predictors with high predictive power may also reduce the set of 19% patients which currently do not receive a prediction resulting in further reducing the amount of work for the clinician.

## 5. Strengths and Limitations of This Study

### 5.1. Strengths

The study’s significant advantage was its extensive sample size of over 827 patients, which was a key benefit for adequately training and testing the models with high accuracy, i.e., the large sample sizes resulted in a very narrow 95% confidence interval of 92% to 96% for model accuracy which was estimated to be 94%. This is important, as it provides an estimation of the accuracy with very high certainty. Another strength was that the model was confronted and independently tested in a very large sample of 249 new, previously unseen patients. This is crucial because a prediction model should empirically demonstrate in a large sample that the model generalizes well to new data with the same model accuracy as in the training sample. It should also highlighted that the model is not only a theoretical concept as we will implement the machine learning model in a real time computer system for practical use by the surgeon to identify patients with low DAP. Another strength concerns the application of the reject option, which considerably increases negative predictive power. This is crucial as we do not want to predict that the patient will receive a low dose of radiation exposure if the patient will actually receive a high dose.

### 5.2. Limitations

DAP was categorized into low and high doses, which was performed because a categorization offers the application of a reject option. On the one hand, by using this method, the level of accuracy in predicting radiation exposure increased significantly, which allows medical professionals to confidently rely on the predictions. On the other hand, the reject option was a limitation of the study, since 19% of all patients did not receive a prediction. If the prediction model cannot be used for certain patients, the clinical expert can still make decisions in their usual manner as part of their daily practice, if they so choose.

## 6. Conclusions

By using the neuronal network model, the radiation exposure of 81% of all study patients undergoing ureterorenoscopy were accurately predicted. In these patients, the model predicted a low DAP with a very high accuracy of 94% (95% CI: 92–96%). The remaining patients in which no predictions were made (19%) were still the most challenging and predictions should be made as per usual in clinical practice. We highlighted several suggestions for future research: firstly, follow-up studies should be made to identify additional predictor variables and to analyse their combined predictive power. This may allow to make correct predictions also for those patients who currently do not receive a prediction. Secondly, we suggest that the definition of intervention-related reference values for radiation exposure needs to be further investigated in this context. Thirdly, we suggest building, training and testing several machine learning models for patients who underwent percutaneous nephrolithotomy if reliable reference values are available.

## Figures and Tables

**Figure 1 jpm-13-00643-f001:**
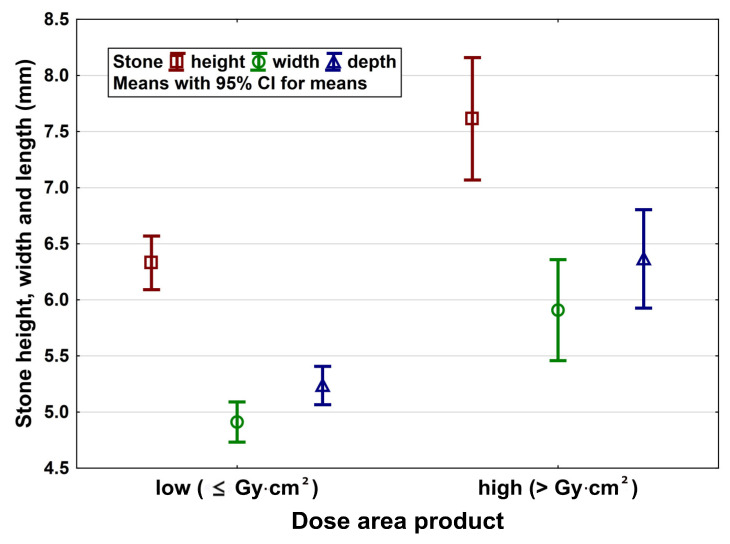
Stone size in width, height and depth differ significantly between patients with low and high dose area product.

**Figure 2 jpm-13-00643-f002:**
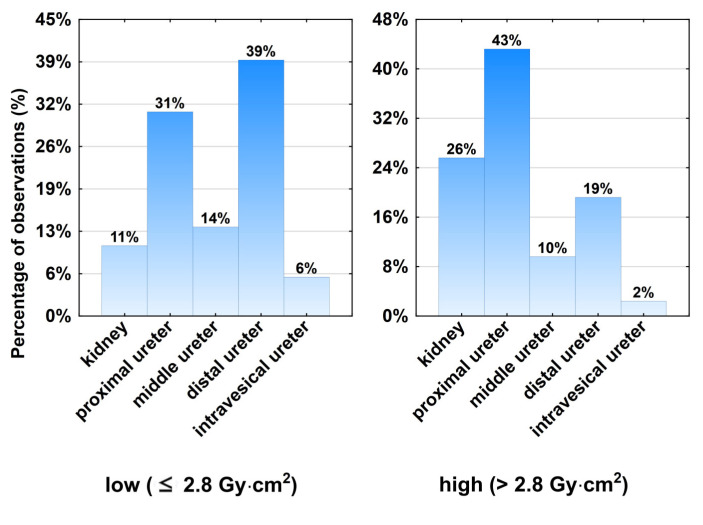
Distribution of stone location on ureterorenoscopy and its relation to lower and increased radiation exposures.

**Figure 3 jpm-13-00643-f003:**
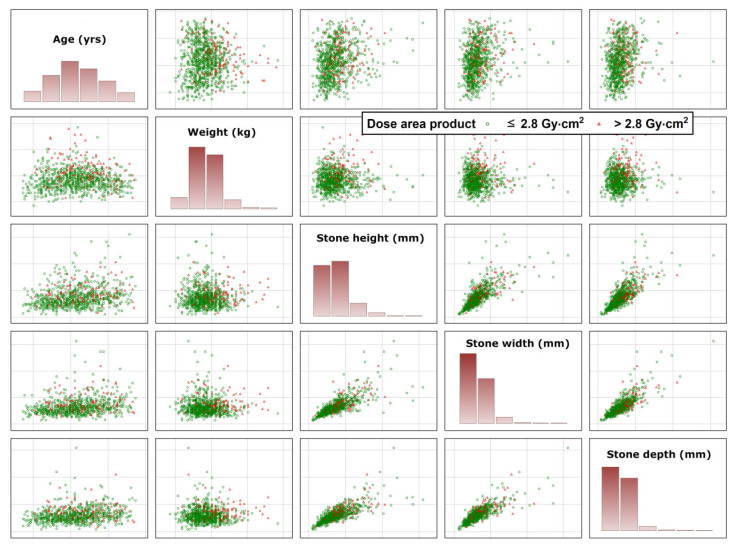
Illustration how close patients with a low DAP and high DAP are stuck together when only some of the predictors without using any reject option are used for separation. Results illustrate a considerable overlap between both groups and the need for application of the reject option. Green circles illustrate patients with low DAP doses, red triangles does with large DAP.

**Table 1 jpm-13-00643-t001:** Patient demographics for URS.

Type of Stone Treatment	URS (*n* = 827)
Gender % (male/female)	67%/33%
Age (mean/std)	53.8/16.1
BMI (kg/m^2^) (mean/std)	27.9/5.5
Dose area product in Gy·cm^2^ (mean/std)	1.5 (2.1)
Ureteral stenting preoperative %	79%
Laserlithotripsy %	24%
Number of stones (mean/std)	1.37 (1.23)
Stone free rate %	92.2%
Use of flexible URS %	31%
Stone location (intrarenal/upper ureter/middle ureter/distal ureter/ureteral orifice) %	(13/33/13/36/5)%
Mean Hounsfield units (mean/std)	503 (155)
Max. Hounsfield units (mean/std)	1061 (411)
Stone height (mm) (mean/std)	6.50/3.30
Stone width (mm) (mean/std)	5.06/2.50
Stone depth (mm) (mean/std)	5.39/2.40

**Table 2 jpm-13-00643-t002:** Results of six machine learning algorithms to predict DAP ≤ 2.8 Gy·cm^2^ and DAP > 2.8 Gy·cm^2^ based on negative (NPV) and positive predictive values (PPV). Results are given before application of reject option and are based on the 10-fold cross-validation.

Model Performance	Gradient Boosted Trees	Support Vector Machine	Nearest Neighbours	Random Forest	Bayes Classifier	Neural Network
(NPV/PPV)%	89/80%	90/89%	86/86%	89/83%	92/57%	93/77%

**Table 3 jpm-13-00643-t003:** Overview of the performance of the best neural network model in predicting dose area product. Results refer after application of the reject option and are given in the training as well as independent test sample to demonstrate its ability to generalize to new previously unseen patients.

	Negative Predictive Power	Positive Predictive Power	Unpredicted	Total Correctly Predicted
Training sample (10-fold cross-validation)	445/472 94%	0/0	106/578 18%	445/472 94%
Independent test sample	186/198 94%	0/0	51/249 20%	186/198 94%
Overall sample	631/670 94%	0/0	157/827 19%	631/670 94%

## Data Availability

The datasets used and/or analysed during the current study are available in anonymised form from the corresponding author on reasonable request.

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
