# Peer review of "Personalized Prediction of Patient Radiation Exposure for Therapy of Urolithiasis: An Application and Comparison of Six Machine Learning Algorithms"

_jpm, 2023, doi:10.3390/jpm13040643_

Round 1

Reviewer 1 Report

dear authors, thanks for sumbmitting your manuscript,

I have a question about your manuscript: you mentioned in your manuscript that patients who underwent semirigid ureteroscopy included in your study, my question is that why did you use fluoroscopy for these patients?

 also in the abstract, you should use complete form of phrases for the first time like DAP 

Author Response

Response to the Reviewer

Manuscript: jpm-2305701

Dear Reviewer,

Thank you for your detailed and helpful review. We have made every effort to meet your requirements and have processed your suggestions. We hope to fulfill all the necessary criteria and would be very pleased with your approval.

Reviewer Comment:

  1. I have a question about your manuscript: you mentioned in your manuscript that patients who underwent semirigid ureteroscopy included in your study, my question is that why did you use fluoroscopy for these patients? Also in the abstract, you should use complete form of phrases for the first time like DAP.

 Author reply: Thank you for your review and your valuable comments. We considered all endourological procedures for stone removal in our work. This also includes semirigid as well as flexible ureterorenoscopies. These procedures are performed in our center at the X-ray workstation with fluoroscopy, as is common practice in most centers in Germany and Europe. Due to the presence of ureteral catheter preoperatively and also due to the possible insertion of a postoperative ureteral catheter, fluoroscopy is necessary in these cases. In addition, fluoroscopy is used in semirigid ureterorenoscopy for diagnostic control, e.g. diagnostic of extravasation or residual stone control. Since this is practiced in many centers, the gain of information about the dose area product is also of interest for this group of patients.

However, we fully agree with you that semirigid ureterorenoscopy can also be performed without fluoroscopy and in these cases there is no radiation exposure at all. This can be decided by the performing urological center. Our work is intended to help better assess personal radiation risk for stone patients, regardless of the approach taken, and we were glad to integrate your point of view into our work.

Author action:  We added the sentence in the introduction:

“There are various arguments to perform certain interventions without fluoroscopy, if they are considered to be an uncomplicated stone removal. This leads accordingly to no radiation exposure [5,6,7]. However, this procedure depends on the center performing the procedure. For example, high-volume clinics with appropriate experience can easily avoid more fluoroscopy for semirigid ureterorenoscopies which may result in simple stone removal. For more complex stone removal or necessary ureteral catheter insertions/removals or control of intraoperative complications, fluoroscopy may be necessary.”

 We added the sentence in the discussion:

 “Another approach is performing endourological stone removal without fluoroscopy. This technique was also shown to be feasible even for PCNL without increased risk to patients or inferior stone clearance free rates [27,28]. However, this is not applicable to all patients and in many centers, the use of fluoroscopy is part of the stone treatment and complication management.”

 In the abstract and the manuscript the complete forms e.g. dose area product are now used as suggested.

I hope we were able to answer your question to your satisfaction and believe that our article has substantially improved with your help. I am looking forward to answer any questions and hope that the revision will find your acceptance.

Reviewer 2 Report

I have reviewed the study titled  ‘’Neural Networks Modeling for prediction of personalized patient radiation exposure during treatment of urolithiasis’’. Machine learning has been a very interesting topic in recent years. The presented study can provide contributions to clinical practice and literature. I have some questions, and I will suggest some revisions.

  1. The data date of the study is 2016–2019; why not get more up-to-date data? The mentioned surgery is one of the most frequently performed surgeries in urology clinics.
  2. The work could be considered valuable, but I think the surgery chosen in the study would be more suitable. Because URS (rigid or flexible) can be done without radiation. For example, flexible URS surgeries in our clinic are almost completely radiation-free. PERCUTANEOUS NEPHROLITHOTOMY would have been a better option than URS.Currently, PNL surgeries are performed with a high rate of fluoroscopy guidance in many centers.
  3. Abstract  should include the time interval of the study.
  4. Introduction: It should be written in three separate paragraphs: first, radiation exposure on URS; second, machine learning; and, in the last paragraph, the purpose of the study.
  5. Materials and methods: should be written with suitable subheadings such as "compliance with ethical standards," "study design," and "ureterorenoscopy technique."
  6. Results: The sentence ‘’ A total of 995 patients undergoing ureterorenoscopy were included into the study 141 between 05/2016 and 12/2019. 1’’ is already writing in the methods section.
  7. Discussion: may include further discussion of the current study.
  8. Conclusion: It's written like a one-to-one repetition of the results section. Instead of this, conclusions, predictions, and ideas for the future should be written.
  9. References: should be updated; there should be at least a few 2022–2023 references. If there is no study on the subject of the study, current references on other general information should be added or changed.

Author Response

Response to the Reviewer

Manuscript: jpm-2305701

Dear Reviewer,

Thank you for your detailed and helpful review. We have made every effort to meet your requirements and have processed all your suggestions. We hope to fulfill all the necessary criteria and would be very pleased with your approval.

Reviewer Comments:

  1. The data date of the study is 2016–2019; why not get more up-to-date data? The mentioned surgery is one of the most frequently performed surgeries in urology clinics.

 Author reply: Thank you for your question and your detailed review. Please note that the sample size is already quite high with 995 patients which were enrolled in the study. This large sample sizes already guarantees a sufficient amount of training, validation and independent test data to end up with stable and statistically sound results.

Additionally, from 2019 - 2022, the impact of the Covid-19 pandemic had an impact on data collection in recent years. Personnel restrictions and home office arrangements led to only limited data acquisition.

However, this had no influence on the data analysis and the results of our study, as the methodology and performance of ureterorenoscopies have not changed significantly from 2020-2022. Our databases are continuously being updated and the implementation of our model in clinical practice is already planned, as well as the conducting of additional prospective studies.

  1. The work could be considered valuable, but I think the surgery chosen in the study would be more suitable. Because URS (rigid or flexible) can be done without radiation. For example, flexible URS surgeries in our clinic are almost completely radiation-free. PERCUTANEOUS NEPHROLITHOTOMY would have been a better option than URS. Currently, PNL surgeries are performed with a high rate of fluoroscopy guidance in many centers.

Author reply: Thank you for your raising this issue. We agree that the fluoroless ureterorenoscopy was not adequately described. Thus, we added various aspects of radiation free surgery to the manuscript. We confirm that patient’s radiation exposure during percutaneous nephrolithotomy were performed with a high rate of fluoroscopy in many centers. Thus, an inclusion of this patient cohort would increase heterogeneity of the patient sample when it comes to radiation levels. This might affect predictive power of the machine learning models.

From a statistical point of view, it is highly likely that heterogeneity of the sample would make it considerably more difficult to end up with accurate predictions for several reasons: First of all, it is more difficult to set up reference values for PCNL. The study we cited and used for the development of our model describe the high variation for DAP (1.6 Gy/cm2 – 30.6 Gy/cm2) as a result of surgeon experience for this specialized procedure. The authors explain this by “surgeons performing the procedure more regularly will use less radiation when gaining access and performing PCNL. It is also worth noting that these high-volume centers receive referrals from other hospitals, often making their case load more complex, making the reduction in DAP even more significant.” [22]. This results particularly for PCNL in radiation exposure that often exceed levels quoted in the literature. The authors quote that therefore it is important to highlight that these reference levels should not be seen as a ‘goal’ for radiation exposure. Nevertheless, we see it as a goal to train and test models for PCNL, preferably with predictions of definitive values that can predict radiation exposure.

Secondly, training and testing of the model does also depend on how many patient underwent URS and how many PNL. In our case, data from less than 10% PNL were available. Even if a good model was found which includes URS and PNL patients (which we think is unlikely), the model should be applied to a sample with the same proportion of URS and PNL. This might reduce practical applicability of the model.

For all these reasons, we think it is a natural way in model building to go ahead step by step. First by training and testing several machine learning models for URS patients and next for PNL patients.

Author action: we added a new section to the introduction and the discussion:

“There are various arguments to perform certain interventions without fluoroscopy, if they are considered to be an uncomplicated stone removal. This leads accordingly to no radiation exposure [5,6,7]. However, this procedure depends on the center performing the procedure. For example, high-volume clinics with appropriate experience can easily avoid more fluoroscopy for semirigid ureterorenoscopies which may result in simple stone removal. For more complex stone removal or necessary ureteral catheter insertions/removals or control of intraoperative complications, fluoroscopy may be necessary.”

“Another approach is performing endourological stone removal without fluoroscopy. This technique was also shown to be feasible even for PCNL without increased risk to patients or inferior stone clearance free rates [27,28]. However, this is not applicable to all patients and in many centers, the use of fluoroscopy is part of the stone treatment and complication management.

For PCNL the stone treatment is usually expected with a higher radiation exposure [29]. Although the development of a prediction model would be useful, the number of patients undergoing PCNL is generally lower than URS patients. Additionally, it is more difficult to find appropriate reference values, as these significantly differ depending on the expertise of the performing center [22]. In addition, there are different conditions due to the type of positioning and technique of the PCNL, which lead to different radiation exposures. Thus, the supine position has an advantage over the prone position with regard to radiation exposure [30]. This decision is up to the surgeon performing the procedure which might enlarge the variations of DAP for PCNL.”

  1. Abstract: Should include the time interval of the study.

Author reply:  Thank you for your comment. Time interval of the study can be found in the abstract. Please note that the time interval of the study is already included in the abstract which reaches from May 2016 to December 2019.

  1. Introduction: It should be written in three separate paragraphs: first, radiation exposure on URS; second, machine learning; and, in the last paragraph, the purpose of the study.

Author reply:  Thank you for your comments on improving the structure of our manuscript. The introduction has now been revised and separate paragraphs have been added as you suggested.

Author action: We have structured the paragraph according to URS and radiation exposure. Then we added the paragraph about machine learning. Finally, the paragraph about the purpose of the study was inserted.

  1. Materials and methods: should be written with suitable subheadings such as "compliance with ethical standards," "study design," and "ureterorenoscopy technique."

Author reply: Also, thank you for your advice to revise the structure of material and methods. Subheadings have now been added now.

Author action: We added subheading 2.1 Compliance with ethical standards. We added subheading ‘2.2 Study design’. In subheading 2.3, the technique of URS is described in detail. Statistical methods section is now given in 2.4.

  1. Results: The sentence ‘’ A total of 995 patients undergoing ureterorenoscopy were included into the study 141 between 05/2016 and 12/2019. 1’’ is already writing in the methods section.

Author reply:  Thank you very much for your comment. The sentence is deleted in the methods section now and belongs to the results section.

Author action: We have deleted the sentence from the methods section. It can be found in the results section.

  1. Discussion: may include further discussion of the current study.

Author reply: Thank you for this important point. We have now expanded the discussion and further addressed the relevance of the model for patients in clinical practice. Additionally, we describe the need to identify additional predictors, as well as the possibilities of obtaining direct radiation exposure with definitive values.

Also, we now discuss the use of fluoroscopy for endourologic stone removal, as well as the use of the model for PCNL as already suggested by you.

Author action: We added the sentences to the discussion:

“Technological advances such as the miniaturization of instruments and the use of effective lasers such as the holmium or thulium laser for lithotripsy allow more and more complex stone diseases to be treated endourologically. With better complication management and improved awareness of pressure and temperature levels on the kidney during endourologic surgery, effective stone decontamination is possible for an increasing number of stone patients through ureterorenoscopy and laserlithotripsy. Considering the in-creasing number of stone patients and thus endourologic stone treatment, radiation exposure is an important issue.”

“Radiation exposure as part of the informed consent process is not only recommended by medical experts, but patients are also becoming more self-aware and are demanding information about the expected treatment procedures and side effects.”

“For patients, the use of our model allows them to better assess their personal radiation risk and to be aware that they do not have an increased radiation exposure. This is a useful asset especially for patients with recurrent stones. The addition of this information in the patient informed consent before surgery increases the quality of the informed consent process and addresses the side effects due to radiation exposure of endourological stone treatment.”

“Another approach is performing endourological stone removal without fluoroscopy. This technique was also shown to be feasible even for PCNL without increased risk to patients or inferior stone free rates [27,28]. However, this is not applicable to all patients and in many centers, the use of fluoroscopy is part of the stone treatment and complication management. For PCNL the stone treatment is usually expected with a higher radiation exposure [29]. Although the development of a prediction model would be useful, the number of patients undergoing PCNL is generally lower than URS patients. Additionally, it is more difficult to find appropriate reference values, as these significantly differ de-pending on the expertise of the performing center [22]. In addition, there are different con-ditions due to the type of positioning and technique of the PCNL, which lead to different radiation exposures. Thus, the supine position has an advantage over the prone position with regard to radiation exposure [30]. This decision is up to the surgeon performing the procedure which might enlarge the variations of DAP for PCNL.”

It would be desirable to directly predict the DAP or entrance skin doses, we suspect that additional predictors are needed for doing so. If these predictors can be gathered in clinical use, then model with these new predictors should be further analyzed for their prediction power and should be used in new machine learning models.”

“The search for new predictors with high predictive power may also reduce the set of  patients which currently do not receive a prediction resulting in further reducing the amount of work for the clinician.”

  1. Conclusion: It's written like a one-to-one repetition of the results section. Instead of this, conclusions, predictions, and ideas for the future should be written.

Author reply: The conclusion is rewritten now and we added three ideas for the future. We still believe that the reader should a receive a summary of the main findings and thus we report in some sentences the most important results.

Author action: we modified the conclusion and the following lines are added now:

”We point out several suggestions for future research: First of all, follow-up studies should be made to identify additional predictor variables and to analyze their combined predictive power. This may allow to make correct predictions also for those patients who currently do not receive a prediction. Secondly, we suggest that the definition of intervention-related reference values for radiation exposure needs to be further investigated in this context. Thirdly, we suggest to build, train and test several machine learning models for patients who underwent percutaneous nephrolithotomy if reliable reference values are available.”

  1. References: should be updated; there should be at least a few 2022–2023 references. If there is no study on the subject of the study, current references on other general information should be added or changed.

Author reply: References were updated with more 2022-2023 references. More studies on machine learning have been added, as well as references describing the use of fluoroscopy. More recent studies from 2022 and 2023 have been added.

Author action: Reference 5-7 was added. Reference 16-21 was added. Reference 27-30 was added.

I hope we were able to answer all your questions to your satisfaction and believe that our article has substantially improved with your help. I am looking forward to answer any questions and hope that the revision will find your acceptance.
